# Analyzing game statistics and career trajectories of female elite junior tennis players: A machine learning approach

**Michal Bozděch** [1]*, **Jiří Zháněl** [2]

1 Department of Physical Education and Social Sciences, Faculty of Sports Studies, Masaryk University, Brno, Czech Republic, 2 Department of Sport Performance and Exercise Testing, Faculty of Sports Studies, Masaryk University, Brno, Czech Republic

☯ These authors contributed equally to this work.

* michal.bozdech@fsps.muni.cz

**Data Availability Statement:** All relevant data are within the paper and its Supporting information files.

## Abstract

Tennis is a popular and complex sport influenced by various factors. Early training increases the risk of career dropout before peak performance. This study analyzed game statistics of World Junior Tennis Final participants (2012–2016), their career paths and it examined how game statistics impact rankings of top 300 female players, aiming to develop an accurate model using percentage-based variables. Descriptive and inferential statistics, including neural networks, were employed. Four machine learning models with categorical predictors and one response were created. Seven models with up to 18 variables and one ordinal (WTA rank) were also developed. Tournament rankings could be predicted using categorical data, but not subsequent professional rankings. Although effects on rankings among top 300 female players were identified, a reliable predictive model using only percentage-based data was not achieved. AI models provided insights into rankings and performance indicators, revealing a lower dropout rate than reported. Participation in elite junior tournaments is crucial for career development and designing training plans in tennis. Further research should explore game statistics, dropout rates, additional variables, and fine-tuning of AI models to improve predictions and understanding of the sport.

## Introduction

### Game statistics in tennis

Tennis can be characterized as a multidirectional, high-intensity intermittent individual or team racket sport played by both men and women (individually or in mixed teams) on various surfaces [1, 2]. It is one of the most popular sports globally, with over 89 million players (1.71% of the population) in 2021, with females accounting for 41% of the players. At the junior level, there were 3,703 registered girls, slightly outnumbering boys (50.9%), but this ratio significantly changed at the professional level, with 59.05% men and 40.95% women [3]. This shift highlights the need for greater focus on the transition from junior to professional categories for girls or addressing the issue of Burnout and Transitions. Success in tennis

**Funding:** JZ was funded by The Masaryk University (Specific research) for the project entitled Laterality in the context of diagnosis of selected factors of sports performance in tennis and injury prevention (Grant No. MUNI/A/1637/2020). The full name of the funder is Masaryk University, and more information can be found on their website: https://www.muni.cz/en/research/projects/59488. The funders had no role in the study design, data collection and analysis, decision to publish, or preparation of the manuscript.

**Competing interests:** The authors have declared that no competing interests exist.

requires a combination of technical and tactical proficiency, mental skills, and high physical performance levels [4–6]. The Association of Tennis Professionals (ATP) and the Women's Tennis Association (WTA) update rankings weekly based on players' points earned in tournaments, determining the top players at the end of the season. Given the complexity of the sport, factors such as endurance, rapid acceleration, deceleration, and powerful strokes influence match outcomes, with matches averaging 1.5 hours or 75 minutes for junior matches at the national level [1, 7]. However, identifying key variables for match, tournament, or season success is challenging due to multiple factors and their varying effects. Artificial intelligence (AI) can help overcome these complexities by appropriately tracking and determining important variables in tennis.

The outcome of a tennis match is influenced by numerous variables that players, opponents, and coaches monitor and strategically utilize. Tools such as video recordings, wearable sensors, players' statistics are analysed be player or opponent [8–10]. However, tracking and determining important variables with the goal of winning a match, tournament, or season is very complicated because it is not easy to determine one decisive factor and different factors tend to have different effects. This can be solved with the use of appropriate AI. Fewer studies have examined the analysis of match statistics in junior tennis, particularly in comparison to professional tennis players. Several researchers have explored these differences [11, 12], with the consensus being that serve and serve return statistics are critical differentiators between junior and professional players.

## Artificial intelligence in sport

Based on the introductory information about tennis, it is evident that the sport involves numerous interrelated associations and covariations that impact the course and outcome of the game. AI has proven to be a suitable approach for predicting future sports events, leveraging its ability to simulate human intelligence [13, 14]. Machine learning (ML), a popular type of AI, is widely used to automate pattern detection in datasets [15]. In the realm of sports, ML algorithms can effectively predict both categorical and metric variables, given appropriate function selection, justification, testing, training, and validation [16]. Specifically, ML serves as a valuable tool for evaluating individual or team performance (serving, return statistics), analyzing career trajectories, identifying tactical patterns (opponent or individual/team), recognizing talented athletes, optimizing training plans for peak performance, and even preventing injuries [13, 17–19].

## Transition from junior to senior age category

The number of theoretical and empirical studies focusing on the junior-to-senior transition (JST), also known as the transition within a career, is increasing, not only in sports-oriented journals [20]. This phase is crucial in the development of young, promising, or elite athletes, as it signifies the shift from junior/youth age categories to the professional level [21]. The JST is characterized by non-linear processes, encompassing a dynamic and complex developmental phase [22–24], along with sociocultural barriers, heightened expectations and standards (emotional, moral, performance-related), and direct demands for mental stability amidst high psychological stress [21, 25, 26]. Athletes often describe this transitional phase, typically lasting one to four years, as the most challenging and critical in their sporting careers. Elite players strive to overcome it with the least risk of injury and negative consequences [21, 25–27]. Consequently, this period contributes to a dropout rate of up to two-thirds among junior athletes [28]. While temporary biological advantages related to birth dates aid in navigating this difficult phase for relatively older peers, there is evidence of a higher dropout rate among athletes

who unknowingly benefited from this advantage, particularly after the end of adolescence and within two years at the professional level, often before reaching their peak performance evident [29–33]. It is important to acknowledge that although external influences may temporarily slow down the consequences of the JST, they cannot be entirely eliminated.

The primary objective of this study encompasses several aspects. Firstly, it seeks to employ AI techniques alongside baseline (non-game) variables to forecast the outcomes of a junior elite tennis tournament. Additionally, it aims to investigate the potential influence of participation and performance in an elite junior tournament on subsequent sports careers. Furthermore, the study aims to discern disparities in game statistics between individuals who did not partake in the junior elite tournament and those who did, specifically within the professional WTA league. Lastly, it aims to explore the feasibility of predicting one's ranking within the professional league by utilizing continuous (percentage-based) and/or cumulative (count-based) game statistics.

## Materials and methods

### Participants

The World Junior Tennis Final (WJTF) tournament, consisting of 16 national teams per year, has been organized by the International Tennis Federation. The participating teams are the winners of the regional round, excluding the host team. Each team is composed of three participants. For this study, particular attention was given to the WJTF participants from 2012 to 2016, as this timeframe allowed the players to reach their peak performance age (around 24 years) during the data collection year [34, 35]. This facilitated a more valid comparison with other professional tennis players. To ensure the accuracy of the study and mitigate potential gender-related influences that could distort the outcomes [27], the research exclusively focused on females. Additionally, player statistics from the top 300 players were included in the study for the purpose of predicting future events, such as ranking, using AI. The research was conducted in accordance with the principles outlined in the Declaration of Helsinki and approved by the Masaryk University Research Ethics Committee (EVK-2021-006).

### Data collection

The research data for the World Junior Tennis Final (WJTF) were obtained from official materials and supplemented with anonymized information provided by the tournament organizers. They were then categorized based on specific research criteria, along with player statistics from the top 300 female players as of the last week in 2022, were collected from the Women's Tennis Association (WTA) database.

### Research data

The research data were divided into three main groups:

1. Official WJTF documents: These included variables such as nomination rank at the tournament, tournament year, players' country, continent, birth year, and final tournament rank.

2. WTA status of the WJTF participants: This included variables related to the participants' registration in the WTA database and their career-best ranking as of November 7, 2022. WTA status was categorized as not found in the WTA database, found but did not accrue any ranking points, and players who had obtained a WTA ranking.

3. Individual player stats from the top 300 female players: This group comprised 19 different variables, categorized into baseline characteristics (rank, age, height, singles matches

played), serving stats (aces, double faults, 1$^{st}$ serve %, 1$^{st}$ serve won, 2$^{nd}$ serve won, break points saved, service points won %, service games won, service games played), and return stats (return points won, 1$^{st}$ return points won %, 2$^{nd}$ return points won %, break points converted, return games won, return games played), as well as their subcategory containing only mentioned percentage variables.

The variable Points was omitted from the analysis due to multicollinearity with the output variable rank. All the data used for analysis in this manuscript are available in S3 Table (Dataset of WJTF and WTA).

## Statistical analysis

Based on an evaluation of the correlation matrix (S4 Fig in Appendix 2), multiple cubic regression functions were calculated to determine the relationships between the selected predictor (independent variable on the x-axis) and response (dependent variable on the y-axis) variables. The cubic regression models were chosen based on the coefficient of determination ($R^2$) to ensure the best fit between the variables.

To predict future events, an artificial intelligence (AI) approach, specifically a machine learning algorithm, was employed. The research dataset, categorized as a classification type of AI model, consisted of 4,835 data points (refer to Table 1) and was randomly divided into training (70%) and testing (30%) sets. The AI models were developed using MATLAB with Machine Learning Toolbox™ (The MathWorks, Inc., v. R2021b, Natick, MA, USA) and IBM SPSS (IBM SPSS Inc., v. 29.0.0 for Windows, Chicago, IL, USA). Cross-validation with a k-fold value of 5 was employed to assess and compare different algorithms. Only the best-performing models from both software platforms were further analyzed and included in this study based on the results of training and testing validation. Both predictors and responses in the dataset were categorical variables. The performance of the models was evaluated using the Area Under the Curve (AUC), which can be interpreted as having no discrimination (AUC < 0.5), acceptable (AUC < 0.8), excellent (AUC < 0.9), or outstanding (AUC > 0.9) [36–38].

Additionally, a separate research dataset was utilized for neural network AI models. This dataset comprised 6,874 data points (refer to Table 2), with the rank being the output variable

**Table 1. Classification AI models for junior elite tennis players: Overview of the best models.**

| No. | Response | Predictor | Classification | | Validation (%) | | AUC |
|---|---|---|---|---|---|---|---|
| | | | Model | Type | Training | Test | |
| [1] | Nomination | Tournament year, Country, Continent, Birth year, Tournament Final Rank | SVM | Coarse Gaussian | 35.9 | 22.9 | 1$^{st}$ = .41 <br> 2$^{nd}$ = .36 <br> 3$^{rd}$ = .37 |
| [2] | Tournament Final Rank | Tournament year, Nomination, Country, Continent, Birth year | Ensemble | Bagged Trees | 75.0 | 87.5 | 1.–4. = .89 <br> 5.–8. = .89 <br> 9.–12. = .84 <br> 13.–16. = .93 |
| [3] | Ranking at WTA | Tournament year, Nomination, Country, Continent, Birth year, Tournament final rank | SVM | Quadratic SVM | 72.9 | 56.2 | Yes = .75 <br> No = .75 |
| [4] | Ranking at WTA (cat.) | Tournament year, Nomination, Country, Continent, Birth year, Tournament final rank | Ensemble | Bagged Trees | 51.3 | 57.7 | 1–500 = .58 <br> 501–1000 = .45 <br> 1001–1500 = .83 |

The AUC variable contains individual categories for the response variable.

**Table 2. Overview of the best neural network AI models for predicting WTA rank.**

| No. | Output | Input variables, n | Stage | MSE | R | Regression |
|---|---|---|---|---|---|---|
| [1] | WTA Rank | ALL, 18 | Training | 539.8788 | .9660 | Output≈0.92*Target+13 |
| | | | Validation | 1.2343e+03 | .9229 | Output≈0.96*Target+12 |
| | | | Test | 2.0787e+03 | .8724 | Output≈1.00*Target+18 |
| [2] | WTA Rank | Baseline characteristic, 3 | Training | 1.0982e+03 | .9287 | Output≈0.86*Target+22 |
| | | | Validation | 557.5054 | .9559 | Output≈0.92*Target+13 |
| | | | Test | 1.4185e+03 | .8892 | Output≈0.84*Target+31 |
| [3] | WTA Rank | Serving stats, 9 | Training | 987.2763 | .9308 | Output≈0.86*Target+21 |
| | | | Validation | 1.4519e+03 | .8892 | Output≈0.85*Target+23 |
| | | | Test | 2.2420e+03 | .8704 | Output≈0.74*Target+40 |
| [4] | WTA Rank | Serving stats_%, 6 | Training | 2.6151e+03 | .8066 | Output≈0.65*Target+55 |
| | | | Validation | 3.2879e+03 | .7871 | Output≈0.58*Target+71 |
| | | | Test | 2.8448e+03 | .7866 | Output≈0.68*Target+53 |
| [5] | WTA Rank | Return stats, 6 | Training | 1.1368e+03 | .9254 | Output≈0.85*Target+27 |
| | | | Validation | 1.2274e+03 | .9125 | Output≈0.83*Target+30 |
| | | | Test | 1.6139e+03 | .8779 | Output≈0.74*Target+45 |
| [6] | WTA Rank | Return stats_%, 5 | Training | 3.3801e+03 | .7468 | Output≈0.55*Target+70 |
| | | | Validation | 2.8029e+03 | .7693 | Output≈0.68*Target+54 |
| | | | Test | 3.6579e+03 | .7249 | Output≈0.58*Target+68 |
| [7] | WTA Rank | Serving + Return stats_%, 11 | Training | 1.7257e+03 | .8865 | Output≈0.74*Target+50 |
| | | | Validation | 3.3595e+03 | .7430 | Output≈0.65*Target+52 |
| | | | Test | 5.0478e+03 | .7098 | Output≈0.72*Target+65 |

The output variable WTA Rank was ordinal, MSE = Mean Squared Error, R = Correlation Coefficient.

(an ordinal variable) and various sets of player statistics as input variables (continuous data; count, percentage). The data for the seven different neural network models were divided into training (70%), validation (15%), and testing (15%) phases, employing the Levenberg-Marquardt backpropagation training algorithm.

An overview of the variables used in Tables 1 and 2 can be found in the Appendix 1; S1 and S2 Tables, respectively. The variables were incrementally applied to the models according to a predetermined methodology based on recommended guidelines. It is important to note that not all available data points were utilized in the study.

Given the violation of assumptions, the Mann-Whitney $U$ test with Bonferroni correction was employed to examine the differences between WTA female players who participated in the junior elite tournament (WJTF 2012–2016) and those who did not. Descriptive comparisons of player statistics were conducted using z-scores, and the results can be found in the supplementary materials (Appendix 2, S1 and S2 Figs). The significance level was set at α = .05 as the threshold for determining statistical significance.

## Results

### Predicting outcomes of junior elite tennis tournaments for girls using non-game characteristics

Predicting the final rank at the junior tournament without considering players' statistics yielded an accuracy of 87.5%, with excellent to outstanding Area Under the Curve (AUC) values (2nd model). This finding highlights the significance of non-game characteristics and

abilities in determining tournament outcomes. It presents an unexpected conclusion for tournament organizers, whose primary goal is to assess, enhance, and compare players' abilities, skills, psyche, and social skills. Conversely, the test accuracy of the 3rd and 4th AI models decreased significantly to 56.2% and 57.7% respectively. This suggests that while irrelevant player statistics can accurately predict junior tournament rankings, they may not be as effective in predicting future career paths. As shown in Table 1, these variables exhibit substantial variability characterized by frequent fluctuations and dynamic changes. The lower accuracy of the 1st model could be attributed to the influence of other variables experienced by all participants in the junior tournament. An overview of the input and output variable types used in the classification AI models can be found in the supplementary material (Appendix 1, S1 Table).

## Transition from junior elite tennis to the professional level for female players

The WTA status was obtained as of December 7, 2022, to track the career progression of the junior elite tennis players. The analysis revealed that out of a total of 240 elite junior players, 38 players (15.83%) were not found in the WTA database, 61 players (25.42%) were found but did not accrue any ranking points, and 141 players (58.75%) had obtained a WTA ranking. Among those who had been ranked ($n = 141$), 59 players achieved a ranking between 1 and 500 (41.84%), 60 players achieved a ranking between 501 and 1000 (42.55%), and 22 players achieved a ranking between 1001 and 1500 (15.60%) as their highest career rank.

Not only did over half (58.75%) of the participants in the elite junior tournament successfully transition to the professional league (WTA), but nearly a quarter (24.58%) achieved a top 500 ranking. This indicates that the participants of the elite junior tournaments have a significant overlap in their transition to professional competition. The fact that even participants from the host country (who were exempt from the elimination prefinals rounds) achieved WTA rankings ($n = 15$; 66.67% were ranked and 26.67% reached the top 500 WTA ranking) supports the conclusion that participation in an elite junior tournament plays a crucial role in shaping future career paths in sports.

Results obtained from the top 300 WTA ranking tennis players in 2022 indicated that 26 players (8.67%) had participated in the elite junior tournament (WJTF from 2012 to 2016), either once or twice.

From Table 1, it is evident that a relatively reliable classification model can be created for predicting rankings in the junior tournament (model 2, with an accuracy of 87.5%). However, despite the large number of successful transfers from junior elite to the professional league, we cannot reliably predict their future success (model 4, with an accuracy of 57.7%) or whether they will make it to the professional league (model 3, with an accuracy of 56.2%) using the same set of variables. This outcome suggests that models 3 and 4 failed to identify a valid trend from the available input variables.

From the total of 302 top 300 tennis players (according to WTA ranking from end of season 2022) 26 (8.61%) players also participated in the finals of the elite tennis tournament (WJTF from 2012 to 2016). To facilitate a comprehensive evaluation of the variations in research variables among the top 300 WTA players, distinguishing those who participated in the WJTF from those who did not, three graphs were generated (Appendix 2, S1–S3 Figs). The analysis revealed notable differences among the variables, particularly in terms of age (with lower z-score values for WJTF participants, as depicted in S1 Fig), 1st serve % (indicating higher z-score values for WJTF participants), 1st serve won (demonstrating higher z-score values for WJTF participants, as shown in S2 Fig), service games won (exhibiting higher z-score values for WJTF participants, as illustrated in S2 Fig), and return games won (displaying higher z-

score values for WJTF participants, as depicted in S3 Fig). Conversely, the z-score values did not indicate substantial differences among the remaining variables. Out of the 18 variables analyzed, WJTF participants demonstrated relatively superior performance in 5 cases (27.78%), excluding age, where younger values were observed, indicating a younger cohort.

To determine if there were differences between the research variables for the top 300 WTA players who did or did not participate in the final of the international junior tournament, the Mann-Whitney $U$ test was utilized due to non-parametric conditions (outliers, non-normal distribution). The results, after adjusting for multiple comparisons using the Bonferroni correction ($\alpha_{adj}$ = .0026), indicated statistically significant differences in only one variable, age ($z$ = -4.854; $U$ = 1526.5; $p < .001$). Players who participated in the WJTF were found to be younger (22.00±1.66 years) compared to other top 300 WTA tennis players (26.76±4.31 years). This suggests that participation in an international tournament may lead to an earlier breakthrough into the top 300 rankings.

Based on the analysis of the participation of players in the elite junior tennis tournament and a comparison with the top 300 WTA players, we present the following recommendations for practical application. Emphasis should be placed on improving the serve as a key element. Clubs, coaches, and investors should consider investing in supporting young talents in junior categories through participation in international junior tournaments. These tournaments can provide players with an advantage by gaining experience, motivation, and enhancing their game strategies. This can help strengthen their skills and prepare them for future success in professional tennis.

## Analysis of the professional tennis player performance and its impact on rankings for female players

Before conducting a detailed data analysis on the prediction of WTA rankings using players' statistics, scatter plot matrices (available in S4 Fig) were initially created. Variables exhibiting a clear relationship were then subjected to further analysis, employing polynomial regression, specifically Cubic regression, as it consistently demonstrated the highest coefficient of determination ($R^2$).

The baseline characteristic sub-criterion (left in S4 Fig) revealed significant associations. Non-linear regression analysis (Cubic) showed that the predictor single matches played explained 87.5% of the variation in points ($F(3,262) = 610.00, p < .001$). Similarly, the predictor single matches played explained 86.3% of the variation in rank ($F(3,262) = 549.32, p < .001$). Another predictor, points, explained 82.4% of the variation in rank ($F(3,298) = 465.70, p < .001$).

In the serving stats sub-criterion (middle in S4 Fig), although lower values of the coefficient of determination ($R^2$) were found, the results were still statistically significant ($p < .05$). The predictor ace explained 75.5% of the variation in double faults ($F(3,263) = 270.63, p < .001$). Based on these results, it can be concluded that female players prefer a more risky/aggressive approach for the first serve, while adopting a more conservative strategy for the second serve. Additionally, the Cubic regression analysis indicated that the predictor ace explained only 42.0% of the variation in 1st serve won ($F(3,262) = 63.31, p < .001$), and 11.2% of the variation in 2nd serve won ($F(3,262) = 10.98, p < .001$). This suggests that top 300 players tend to play it safe on the second serve.

In the return stats sub-criterion (right in S4 Fig), the predictor return points won (%) explained 75.1% of the variation in 1st return points won (%) ($F(3,263) = 397.07, p < .001$). This highlights the importance of returns from the first serve in obtaining a game point. Other associations were either not significant or affected by high multicollinearity. By summarizing

these findings, it becomes evident that single matches played, points, ace, and return points won (%) are influential factors in determining players' rankings and performance in various aspects of the game. A detailed overview of the variables used in Table 2 can be found in Appendix 1 S2 Table.

Model 2 exhibited the highest test accuracy of 79.07%. While it may appear that age is the decisive variable for Model 2, as indicated in Table 2 (which contains three input variables), analysis of S4 Fig reveals that the most significant input variable for this model is the number of singles matches played, as mentioned previously. This relationship can be expressed as follows: the more matches a player participates in, the better (lower) ranking they achieve. For instance, consider the case of Iga Swiatek, the top-ranked female player. She played 74 matches in the season, accumulating a total of 10,335 points, resulting in an average of 139.66 points per match. In comparison, the average for the top 300 players is 508.72±915.88 points, with an average of 16.79±17.41 matches played or 30.81±28.41 points per match. Ons Jabeur, the second-best player in the 2022 season, played 64 matches and earned 4,555 points, equating to an average of 71.17 points per match. Conversely, the lowest-ranked player in this study within the top 300 (Zoe Hives) played only one match and obtained 20 points. Considering that Iga Swiatek was only 21 years old, whereas the average age of the top 300 players is 25.66±4.50, one can expect significant success in her sports career. Considering her participation in the WJTF tournament twice (with her team reaching the quarterfinals in 2014 and the finals in 2015), it can be theoretically inferred that her involvement in the tournament might have contributed to her potential for significant success in her sports career.

## Discussion

The research plan of this study can be divided into 4 main objectives: (a) forecasting the outcomes of a junior elite tennis tournament using AI; (b) investigating the potential influence of participation and performance in an elite junior tournament on subsequent sports careers; (c) discerning disparities in-game statistics between senior WTA players who participate and do not participate in the junior elite tournament; (d) predicting WTA ranking using AI.

In the case of predicting junior elite tennis outcomes from girls' tournaments using only non-game characteristics, the results suggest that while irrelevant player statistics can accurately predict junior tournament rankings, they may not be as effective in predicting future career paths. However, it is crucial to recognize that participating in a junior elite tournament can contribute to increased self-confidence, internal motivation, valuable experience, personal connections, and visibility among relevant stakeholders such as clubs, organizations, coaches, scouts, and potential investors [21, 25, 26]. Furthermore, the lower accuracy of the first model from Table 1 (nomination rank at the tournament as the response variable) could be attributed to the influence of other variables experienced by all participants in the junior tournament. For instance, the relative age of participants, which holds significant weight at this performance level and age category [29, 30], might have introduced bias in the research sample. The insights gained from this part of the study have the potential to be crucial for practical implementation in the tennis industry, particularly for tournament organizers, coaches, and investors. Specifically, these findings can assist in optimizing training programs for junior players. Developing non-game characteristics as one of the main key elements can indirectly influence players' performance and prepare them for a more likely successful professional career. Organizations and clubs, as well as coaches, can rework their talent evaluation criteria based on these findings. Lastly, it is essential to recognize that participation in junior tournaments can foster better collaboration among players, clubs, coaches, scouts, and organizers of

international tournaments. Establishing a strong network and cooperation can benefit the entire tennis community.

Results from the transition of female junior elite tennis players to the professional level indicate that participants in elite junior tournaments have a significant overlap in their transition to the professional competition level. Consequently, the potential negative consequences of the junior-to-senior transition may not have the same magnitude as in other sports that have distinct demands in terms of transition perception, financial support, minimum psychological and mental requirements, and other factors [20]. The fact that even participants from the host country (who were exempt from the elimination prefinals rounds) achieved a superlative level in the WTA rankings ($n = 15$; 66.67% were ranked, and 26.67% reached the top 500 WTA ranking) supports the conclusion that participation in an elite junior tournament plays a crucial role in shaping future career paths in sports. This finding suggests that factors beyond performance, anthropometric characteristics, abilities, skills, and health contribute to the development of sports careers. More precisely, experience also plays an important role. It is important to note that even though the team from the host country did not have to go through the elimination rounds, it does not mean that it would not reach the finals, as there were also female players in the teams who achieved excellent results at the professional level. Ten out of fifteen hosting players (from 2012–2016 WJTF tournaments) achieved rankings between 501–1000 (exactly 6 players) and within the top 500 (exactly 4 players; the best performance was by Marketa Vondrousova with a 14th WTA Rank) as their best career rank. Two players (Marie Bouzkova at 26th rank, Marketa Vondrousova at 85th rank) were among the top 300 WTA players in the 2022 season. However, we cannot unequivocally determine a causal relationship. However, according to the results obtained from the top 300 WTA ranking tennis players in 2022, it was found that 26 players (8.67%) had participated in the elite junior tournament (WJTF from 2012 to 2016). It is crucial to acknowledge that these figures might have been influenced by factors such as early termination of a sports career, possibly due to injury, burnout, family-related interruptions, or pregnancy [31, 32]. However, it is important to note that these variables were not analyzed as part of the scope of this research.

The outcomes from Table 1 suggest that models 3 and 4 (predicting status and categorical level at the WTA ranking) failed to identify a valid trend from the available input (non-game) variables. To enhance the predictive accuracy of the model, additional valid variables could be incorporated, including scale variables, while eliminating variables with low importance [39, 40]. In practice, this could involve expanding the dataset to include data from multiple years of the junior tournament and analyzing additional factors such as player height, arm length, preferred playing style (defensive or offensive, at the net or baseline), experience (number of matches or participation in international tournaments), results of strength assessments, psychological tests, type of court, and player statistics related to serving and returning [6, 41–45], since these are generally recognized variables that influence the outcomes of tennis matches.

The study analysis revealed notable differences among the variables, particularly in variables such as age, 1st serve %, 1st serve won, service games won, and return games won. The importance of serving in tennis is not only logical but has also been addressed in the first publication dedicated to the statistical analysis of tennis [46]. The authors further added that if opponents have similar serving quality, it can be assumed that the match will last longer. However, a more recent study by Cui et al. [47] found that the importance of the serve (and running variables) is more likely to decrease, while return-based variables increase in importance between the first and last sets. Service is, therefore, an important variable, but its importance is not constant during a single match.

WJTF participants reveal relatively superior performance in 5 of the 18 research variables (27.78%), excluding age, where younger values were observed, indicating a younger cohort.

These outcomes may have been influenced by their participation in the junior elite tournament, where they gained valuable experience, motivation, and potentially refined their game strategies, tactics, psychological factors, and other aspects of their gameplay [48, 49], as stated above. In a study conducted by Li et al. [50], it was observed that elite professional players dedicated approximately 10 years of training to reach the international junior level, followed by an additional 10 years to achieve their career peak ranking. However, a more recent study by Li et al. [51] yielded contrasting findings, indicating that the age at which players commenced playing tennis did not significantly impact their peak career ranking. Nevertheless, the combination of age and early ranking emerged as a more reliable predictor than individual variables alone. The identification of influences suggesting the transfer of experience gained at the elite junior level to the senior professional category underscores the significance of these findings. However, we found a significant influence of age in the top 300 WTA players. Therefore, we lean towards the age-specificity of a long-term training program, which during adolescence could focus on stroke techniques and agility, due to the great trainability, which could subsequently be followed by lower-extremity muscle power and upper-extremity muscle strength training in adulthood [52]. It is important to note that the analysis was limited to the top 300 female tennis players, which introduces a potential bias. To enhance the generalizability of future research, it would be advisable to expand the research sample. By including a broader range of players across different rankings, a more comprehensive understanding of the transferability of junior elite experience to the professional level can be achieved.

Our findings suggest that participation in an international tournament may lead to an earlier breakthrough into the top 300 WTA rankings. This finding aligns with the study conducted by Kovalchik et al. [53], which also concluded that younger players achieved better rankings. However, no statistically significant differences were found in other game statistics or variables. It is likely that these non-significant differences can be attributed to accumulated experience, skills, mental resilience, and other factors that may contribute to overall performance.

By summarizing these findings, it becomes evident that single matches played, points, aces, and return points won (%) are influential factors in determining players' rankings and performance in various aspects of the game. In cases where there is an obscure relationship, utilizing AI is appropriate, as it enables the discovery of hidden connections between variables [13, 15]. For these reasons, one of the objectives of the study was to develop an AI model that would enable athletes and their coaches to predict their position in the WTA rankings based on their ongoing player statistics. However, the models incorporating only the percentage variables (models 4–6 from Table 2) exhibited the lowest level of reliability ($R^2$) with values of 61.87%, 52.55%, and 50.38%, respectively. A higher level of reliability would indicate that the model serves as a valuable supplementary source of information for enhancing both long-term and short-term training plans [52]. In the case of tennis, this level of accuracy is considered ambiguous [13]. The rationale behind the decision to develop a highly accurate model utilizing variables based on percentages was rooted in the belief that percentages, such as % ace per match, would offer more consistent and reliable information throughout the season when compared to count numbers (e.g., the number of aces per season). Count numbers were expected to gain accuracy as the season progressed. However, this objective was not adequately attained, and as a result, a sufficiently reliable AI model was not constructed for practical use during the season.

## Limitations and directions for future research

In this study, we conducted a power analysis using a predetermined value of 1-$\beta$ equal to 0.95. Based on the current effect, we found that to achieve this sufficient power, our sample

needed to include 7 female tennis players from the WJTF organization and only 73 players from the WTA organization. This information suggests that our study was able to detect a relevant effect with a high probability, with the actual value of 1-$\beta$ for the statistically significant variable, age, being 0.999. This information strengthens the credibility of our results and supports the validity of our conclusions within the context of our research. However, the study includes a sufficiently large sample (WJTF = 240, WTA = 302) for the defined research objectives with logically chosen variables (6 WJTF-related variables and 21 WTA-related variables), and their quantity (1440 WJTF-related data points and 6089 WTA-related data points). Even though the research still has its limitations, constraints, and shortcomings. For example, it's crucial to acknowledge certain limitations inherent to its scope. While the primary objective involves integrating AI techniques with baseline (non-game) variables to predict the outcomes of junior elite tennis tournaments, there are constraints associated with the availability and accuracy of historical data. Additionally, exploring the influence of participation and performance in elite junior tournaments on subsequent sports careers may face challenges in establishing direct causation due to various external factors, which were not analyzed in this study. These factors could include health status, both acute and chronic injuries, participants' regional backgrounds, the level of support from stakeholders (family, friends, and others), psychological and mental resilience of adolescent participants, social and economic circumstances, coaching strategies and tactics, chance occurrences, and random events. Furthermore, the study aiming to identify disparities in game statistics between individuals who participated in junior elite tournaments and those who did not in the professional WTA league is contingent upon the comprehensiveness and reliability of the available datasets. Lastly, the study's endeavor to assess the feasibility of predicting professional league ranking using continuous (percentage-based) and/or cumulative (count-based) game statistics requires careful consideration of the inherent complexities involved in ranking systems and player performance dynamics. Professional tennis is a multifaceted sport where numerous variables need to be taken into consideration, including the player's and opponents' (current and possibly subsequent) condition (fatigue, psychological and mental resilience, acute or chronic injuries, etc.) and capabilities (technical skills, physical endurance, tactical intelligence, mental strength, reaction speed, anticipating opponent's moves, understanding the opponent's strategy, etc.) [4, 43, 47, 52, 54, 55]. Despite all research variables included in this study, it is evident that defining the main variables in tennis poses considerable challenges. Achieving a precise delineation of these variables proves to be a complex task, given the multifaceted nature of the sport. While it is challenging to establish definitive boundaries for these variables, the intricate interplay of various factors highlights the nuanced and evolving nature of tennis-related parameters. Thus, arriving at a comprehensive understanding of the principal variables in tennis requires a nuanced approach that acknowledges the dynamic and multifactorial aspects inherent in the sport.

This study focused solely on junior and professional senior female tennis players. It did not include a male sample due to significant disparities in performance between sexes at the junior and professional senior levels [1, 12, 56]. Even if it were possible (probably with sufficient accuracy based on the results from the literature) to predict gender and age level, the practical applicability of this model would be minimal. Since the error rate of this model would negatively affect the results of the current study, we limited our focus to female tennis players. This decision was also influenced by the absence of similar studies targeting girls/women. Analyzing and interpreting the trajectories and performances of men and women would have significantly lengthened the manuscript. Additionally, differences in variables at the official ATP and WTA websites would present problems for data comparison.

## Conclusions

The outcomes of the tournament can be predicted with a satisfactory level of accuracy (87.5%); however, utilizing selected (categorical) variables alone does not provide enough precision to determine an athlete's future sports career. This indicates that the ranking within an elite junior tournament is influenced by distinct phenomena/variables compared to those governing the career trajectory. Consequently, it is essential to consider these factors when devising both short-term training plans, such as preparation for an elite junior tournament, as well as long-term plans for a sports career.

Although nearly a quarter of the participants in the elite junior tournament achieved a top 500 ranking in their professional sports career, this study did not identify the most crucial variables or their combination for accurately predicting future tennis careers. To enhance accuracy in career prediction, it would be advisable to expand the scope of research by including a larger number of junior tournaments spanning multiple years and incorporating more relevant and metric variables. Additionally, it would be beneficial to encompass a broader representation of female tennis players within the WTA ranking, ideally including all tournament participants.

Based on the findings, it can be concluded that participation in an elite junior tournament is a significant variable for the future career development of tennis players. This is supported by the observation that a majority of participants from the host country achieved rankings in the WTA. Therefore, the involvement in junior elite tournaments holds crucial importance for the future development of female tennis players.

Based on the data presented, it can be theorized that the actual dropout rate in tennis is lower than what has been reported in the literature. This hypothesis is supported by the observation that a significant number of players were found and ranked in the WTA, indicating their continued involvement and commitment to the sport. Furthermore, the distribution of rankings suggests a gradual progression, with a notable portion of players achieving ranks within the top 500. This theory implies that as players demonstrate greater skill and dedication, their likelihood of persisting in the sport and achieving higher rankings increases.

Statistically significant distinctions were found between the top 300 WTA players who participated in the junior elite tournament and those who did not, particularly in terms of age. The junior tournament participants were significantly younger, suggesting that their involvement in the junior tournament could potentially contribute to an earlier breakthrough into the top 300 rankings.

Drawing upon various insightful findings from the statistics of professional top 300 WTA players (such as the relationship between 1st serve and aces, but not 2nd serve and aces), multiple machine learning models were developed to ascertain the impact of player statistics on WTA rankings. Nonetheless, a sufficiently accurate AI model that exclusively employs percentages as input variables was not constructed. Nevertheless, the resulting test accuracy was substantial enough to validate the viability of utilizing such models. Nevertheless, further refinement is required for the selected models, and additional data, along with relevant variables, must be acquired to enhance model accuracy.

Overall, this research highlights the complexity of predicting future tennis careers and emphasizes the need for continuous data analysis, model refinement, and the inclusion of comprehensive variables to enhance accuracy in forecasting rankings and career paths in the sport of tennis.

## Supporting information

**S1 Fig. Difference of standardized baseline characteristics of the top 300 ranked female tennis players.**
(TIF)

**S2 Fig. Difference of standardized serving stats of the top 300 ranked female tennis players.**
(TIF)

**S3 Fig. Difference of standardized return stats of the top 300 ranked female tennis players.**
(TIF)

**S4 Fig. Scatter plot matrix of baseline characteristics (left), serving stats (Middle), return stats (Right).**
(TIF)

**S1 Table. An overview of the input and output type and use in classification models.**
(DOCX)

**S2 Table. Overview of used variables for the neural network.**
(DOCX)

**S3 Table. Dataset of WJTF and WTA.**
(XLSX)

## Acknowledgments

We gratefully acknowledge the organizers of the World Junior Tennis Final for providing non-public data for this research.

## Author Contributions

**Conceptualization:** Michal Bozděch, Jiří Zháněl.

**Data curation:** Michal Bozděch.

**Formal analysis:** Michal Bozděch.

**Methodology:** Michal Bozděch, Jiří Zháněl.

**Supervision:** Michal Bozděch.

**Validation:** Michal Bozděch, Jiří Zháněl.

**Visualization:** Michal Bozděch.

**Writing – original draft:** Michal Bozděch.

**Writing – review & editing:** Jiří Zháněl.

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
