## [Decision Letter · Decision Letter 0]

12 Oct 2023

PONE-D-23-24174Analyzing Game Statistics and Career Trajectories of Elite Junior Tennis Players: A Machine Learning ApproachPLOS ONE

Dear Dr. Bozděch,

Thank you for submitting your manuscript to PLOS ONE. After careful consideration, we feel that it has merit but does not fully meet PLOS ONE’s publication criteria as it currently stands. Therefore, we invite you to submit a revised version of the manuscript that addresses the points raised during the review process.

We look forward to receiving your revised manuscript.

Kind regards,

Ersan Arslan, Ph.D.

Academic Editor

PLOS ONE

Journal Requirements:

Reviewers' comments:

Reviewer's Responses to Questions

**Comments to the Author**

1. Is the manuscript technically sound, and do the data support the conclusions?

Reviewer #1: Yes

Reviewer #2: Yes

2. Has the statistical analysis been performed appropriately and rigorously? 

Reviewer #1: Yes

Reviewer #2: Yes

3. Have the authors made all data underlying the findings in their manuscript fully available?

Reviewer #1: Yes

Reviewer #2: No

4. Is the manuscript presented in an intelligible fashion and written in standard English?

Reviewer #1: Yes

Reviewer #2: Yes

5. Review Comments to the Author

Reviewer #1: The reviewer would like to thank the authors for choosing this very interesting topic within the scientific field of player development, and more importantly, junior to senior transition in tennis.

In addition, the reviewer considers that he is quite familiar with this field of study in tennis and that he has a particular interest in new study methods and proposals in order to make them more useful for all people involved in tennis but especially for the coaches and experts in charge of long-term player development pathways.

Therefore, the reviewer has read with interest and curiosity that the manuscript that is submitted for consideration and understands that it is a novel contribution to this field of research that is so relevant to tennis.

Regarding the introduction, the reviewer considers that the topic of study is suitably justified and the appropriate and convenient bibliographical references are provided to develop a theoretical framework that allows understanding the importance of the research that was carried out.

On the other hand, the objectives proposed in the research are equally appropriate and also innovative since they attempt to study aspects that until now had not been covered by researchers. Notably, the use of AI is a very interesting aspect of the study.

Regarding the methodological aspects, the reviewer wants to highlight the importance of the study sample because it is not easy to access this type of data in quality research. The reviewer considers that the statistical analyzes carried out are appropriate and convenient for this type of study and therefore the methodological aspects of this work are considerably well treated.

Perhaps in the title of the paper, and the rest of the descriptions, since the sample is composed by female players, this should be included to provide a better understanding of the scope of the study.

The presentation of the results in terms of the figures and tables that are attached is appropriate as it allows the reader to understand in greater depth the narrative that accompanies the graphic elements of the manuscript.

Regarding the discussion, which is presented together with the results, the reviewer considers that it is well structured and also tries to compare the results obtained with those that have been carried out previously so that a complete overview of the state of the research is presented.

Regarding the conclusions section, the reviewer would suggest that the author include a paragraph related to the limitations of this study because it is always important to note those aspects that could be improved in any research process.

Likewise, the reviewer has missed a broad section on the practical applications of the present study because although the introduction discusses the relevance of junior to senior transition for the work of coaches, perhaps it would be advisable to include practical proposals that coaches can consider derived from of the results of this research.

Finally, the reviewer once again wants to thank the authors for their excellent work and ask them to consider the comments she has made as suggestions and indications that are part of the academic review process of a scientific article.

Reviewer #2: This study analyzed game statistics of World Junior Tennis Final participants from 2012-2016 and their career trajectories. It also investigated the impact of game statistics on the rankings of the top 300 female players, aiming to create an accurate model using percentage-based variables. Various statistical methods, including neural networks, were applied, resulting in several machine learning models. While tournament rankings could be predicted using categorical data, subsequent professional rankings could not. The study identified effects on rankings among top 300 female players but couldn't establish a reliable predictive model using only percentage-based data. AI models provided insights into rankings and performance indicators, revealing a lower dropout rate than previously reported. Participation in elite junior tournaments was found to be crucial for career development and training plan design. Further research should delve into game statistics, dropout rates, additional variables, and AI model refinement to enhance predictions and understanding of the sport.

The paper is intriguing, focusing on an innovative topic: assessing the potential contribution of AI and ML in predicting an athlete's career progression (though, it is reassuring that AI still has much to learn before "replacing" the expertise of a team of coaches). Technically, the paper is well-executed. The introduction is well-structured, and the purpose is clear. I would ask the authors to provide a more detailed explanation for their focus solely on women and the exclusion of men (recognizing the general lack of women-focused studies, this should be further justified). The methods are clear and well-described. However, the decision to incorporate results and discussion within the same paragraphs is stylistically questionable. It might be more effective to present the results first and then discuss them separately. Additionally, there are some typographical errors in the text, suggesting a thorough proofreading (preferably by a native speaker) is needed.

6. PLOS authors have the option to publish the peer review history of their article (what does this mean?). If published, this will include your full peer review and any attached files.

Reviewer #1: **Yes: **Miguel Crespo

Reviewer #2: No

---

## [Author Response · Author response to Decision Letter 0]

23 Oct 2023

Dear reviewers and editor,

We would like to express our sincere gratitude for your time, patience, and meticulous analysis of our article. Your comments were invaluable and contributed significantly to improving the quality of our work. Your constructive criticism allowed us to see our article from a new perspective, and we made necessary revisions that wouldn't have been possible without your valuable insight.

We want to emphasize that in our document "Respond to reviewers," we endeavored to address each of your comments and opinions. Your feedback helped us understand the weak points of our work and elevate it to a higher standard. We are thankful for your expert input, which allowed us to substantially enhance our article.

Once again, thank you for your support and dedication. 

With respect, Authors.

---

## [Editor Report · Decision Letter 1]

30 Oct 2023

PONE-D-23-24174R1Analyzing game statistics and career trajectories of female elite junior tennis players: a machine learning approachPLOS ONE

Dear Dr. Bozděch,

Thank you for submitting your manuscript to PLOS ONE. After careful consideration, we feel that it has merit but does not fully meet PLOS ONE’s publication criteria as it currently stands. Therefore, we invite you to submit a revised version of the manuscript that addresses the points raised during the review process.

We look forward to receiving your revised manuscript.

Kind regards,

Ersan Arslan, Ph.D.

Academic Editor

PLOS ONE

Journal Requirements:

Additional Editor Comments:

Dear Authors

This study analyzed game statistics of World Junior Tennis Final participants from 2012-2016 and their career trajectories. It also investigated the impact of game statistics on the rankings of the top 300 female players, aiming to create an accurate model using percentage-based variables. Various statistical methods, including neural networks, were applied, resulting in several machine learning models. While tournament rankings could be predicted using categorical data, subsequent professional rankings could not. The study identified effects on rankings among top 300 female players but couldn't establish a reliable predictive model using only percentage-based data. AI models provided insights into rankings and performance indicators, revealing a lower dropout rate than previously reported. Participation in elite junior tournaments was found to be crucial for career development and training plan design. Further research should delve into game statistics, dropout rates, additional variables, and AI model refinement to enhance predictions and understanding of the sport.

The paper is intriguing, focusing on an innovative topic: assessing the potential contribution of AI and ML in predicting an athlete's career progression (though, it is reassuring that AI still has much to learn before "replacing" the expertise of a team of coaches). Technically, the paper is well-executed. The introduction is well-structured, and the purpose is clear. I would ask the authors to provide a more detailed explanation for their focus solely on women and the exclusion of men (recognizing the general lack of women-focused studies, this should be further justified). The methods are clear and well-described. However, the decision to incorporate results and discussion within the same paragraphs is stylistically questionable. It might be more effective to present the results first and then discuss them separately. Additionally, there are some typographical errors in the text, suggesting a thorough proofreading (preferably by a native speaker) is needed.

---

## [Author Response · Author response to Decision Letter 1]

10 Nov 2023

Journal Requirements:

review your reference list to ensure that it is complete and correct

• We have checked the references list, and it is complete and correct.

Additional Editor Comments:

Technically, the paper is well-executed. The introduction is well-structured, and the purpose is clear. 

• Thank you for your positive feedback on our work.

I would ask the authors to provide a more detailed explanation for their focus solely on women and the exclusion of men (recognizing the general lack of women-focused studies, this should be further justified). 

• In the manuscript, they expanded and added the following justification for why the study focuses solely on a female sample:

o „This study focused solely on junior and professional senior female tennis players. It did not include a male sample due to significant disparities in performance between sexes at the junior and professional senior levels [1,12,56]. Even if it were possible (probably with sufficient accuracy based on the results from the literature) to predict gender and age level, the practical applicability of this model would be minimal. Since the error rate of this model would negatively affect the results of the current study, we limited our focus to female tennis players. This decision was also influenced by the absence of similar studies targeting girls/women. Analyzing and interpreting the trajectories and performances of men and women would have significantly lengthened the manuscript. Additionally, differences in variables at the official ATP and WTA websites would present problems for data comparison. “

• Thank you for this comment, and we hope it will serve as sufficient justification.

The methods are clear and well-described. 

• Thank you for the positive and appreciated feedback.

However, the decision to incorporate results and discussion within the same paragraphs is stylistically questionable. It might be more effective to present the results first and then discuss them separately. 

• Based on this recommendation, we divided this chapter into two separate chapters. We deleted the relevant text from the results and reformatted it into the discussion chapter, to which we added new citations. Specifically, there were 10 citations added.

Additionally, there are some typographical errors in the text, suggesting a thorough proofreading (preferably by a native speaker) is needed.

• We have ensured proofreading and corrected typos in the text.

---

## [Editor Report · Decision Letter 2]

15 Nov 2023

Analyzing game statistics and career trajectories of female elite junior tennis players: a machine learning approach

PONE-D-23-24174R2

Dear Dr. Bozděch,

We’re pleased to inform you that your manuscript has been judged scientifically suitable for publication and will be formally accepted for publication once it meets all outstanding technical requirements.

Kind regards,

Ersan Arslan, Ph.D.

Academic Editor

PLOS ONE
---

## [Editor Report · Acceptance letter]

20 Nov 2023

PONE-D-23-24174R2 

Analyzing game statistics and career trajectories of female elite junior tennis players: a machine learning approach 

Dear Dr. Bozděch:

I'm pleased to inform you that your manuscript has been deemed suitable for publication in PLOS ONE. Congratulations! Your manuscript is now with our production department. 

Kind regards, 

on behalf of

Dr. Ersan Arslan 

Academic Editor

PLOS ONE